# The Role of Nursing in the Management of Chemotherapy Extravasation: A Systematic Review Regarding Public Health

**DOI:** 10.3390/healthcare12141456

**Published:** 2024-07-22

**Authors:** Antonio Antúnez-Blancat, Francisco-Javier Gago-Valiente, Juan-Jesús García-Iglesias, Dolores Merino-Navarro

**Affiliations:** 1Department of Nursing, Faculty of Nursing, University of Huelva, 21007 Huelva, Spain; antonio.antunez@alu.uhu.es (A.A.-B.); lola.merino@denf.uhu.es (D.M.-N.); 2Centro de Investigación en Pensamiento Contemporáneo e Innovación para el Desarrollo Social (COIDESO), University of Huelva, 21007 Huelva, Spain; 3Preventive Medicine and Public Health Area, Department of Sociology, Social Work and Public Health, Faculty of Work Science, University of Huelva, 21007 Huelva, Spain; juanjesus.garcia@dstso.uhu.es

**Keywords:** chemotherapy, extravasation, prevention, nursing, public health, healthcare

## Abstract

The scientific literature was reviewed with the aim of analysing the state of the art on the role of nursing in the management of chemotherapy extravasation, recognising the possible risk factors and identifying effective training programmes for nurses. WOS, Scopus, and PubMed databases were used to perform the searches. Papers that met the inclusion criteria and that had been published in the last 9 years were selected. The Effective Public Health Practice Project (EPHPP) instrument was applied to the selected studies. In addition, this research was registered in the International Prospective Register of Systematic Reviews (PROSPERO) (ID: 512480). Out of the 23 initially selected articles, a total of 9 articles were eventually included, since they met the eligibility criteria that were established after a more exhaustive analysis, which included reading their abstracts and full texts. The results show that the management of chemotherapy extravasation is closely related to factors that largely depend on the nursing staff. Among the most relevant findings are factors including the identification of the nursing role in the management of extravasation due to chemotherapy; risk factors; and effective training programmes for nursing. Nurses play a crucial role throughout the entire process of treatment, prevention, and health education in chemotherapy treatment. Training programmes for nurses are fundamental, as they increase their professional competence and improve the safety of the patient. Adequate knowledge of chemotherapy treatment and the risk factors of extravasation are basic elements for the prevention of this type of injury, as well as for the improvement of the quality of life of patients under this kind of intravenous therapy.

## 1. Introduction

Currently, cancer is one of the main causes of death in the world and a major public health problem. According to the World Health Organisation, it accounted for nearly 10 million deaths in the year 2020, representing one sixth of the total number of deaths recorded in the said year worldwide [1].

In Spain, based on the most recent data of the Spanish Society of Medical Oncology (SEOM), in the year 2022, the cancer rate was 280,110 cases, resulting in a slight increase compared to the previous years. It is important to highlight the mortality rate that took place, with nearly 113,000 deaths [2]. Due to this increase in cancer rate, there was also an increase in the use of the different available treatments, with the most widely used being intravenous chemotherapy, in which different types of antineoplastic agents are employed; these agents act on cancer cell growth, slowing it down or even halting it [3].

Antineoplastic agents are among the modalities that are most widely used to treat cancer. They can be defined as chemotherapeutic drugs that are employed to treat cancer. Their main action is based on removing cancerous cells by inhibiting their division, thereby interrupting their cell cycle and preventing the spread of the disease. However, these agents cannot distinguish healthy cells from cancerous cells, thus they will act on both, thereby resulting in adverse effects [4]. Extravasation is one of such effects, and it consists of the accidental leak of antineoplastic drugs during the venous infusion procedure outside of the vein of choice. This can be due to different intrinsic factors of the blood vessel itself, caused by the displacement of the cannula outside of the blood vessel or inadequate placement. It is one of the most severe complications of the intravenous administration of chemotherapy [5]. Extravasation includes a range of consequences, from simple pain or inflammation in the infusion area to tissue necrosis, with a permanent function loss in the affected limb, which is considered a medical emergency. This situation causes a delay in the treatment and distrust in the patients [6]. With this type of accident, the patient could have serious clinical implications, such as increased morbidity, extended hospitalisation, and alterations in the quality of life [6].

The rate of extravasation is between 0.1% and 6% in the case of peripheral administration, and between 0.26% and 4.7% when the infusions are administered through a central venous access [7]. These extravasation rates are associated with different risk factors: those that are related to the patient (such as age or general health); those related to the product (such as its irritant capacity or toxicity); and those related to the procedure (such as the puncture technique or the administration of the medication), the latter being the most important to take into account [8]. Within the administration process of intravenous chemotherapy, the nurse plays a relevant role in important aspects like patient preparation, drug administration, monitoring of adverse effects, management of documents, and collaboration with the medical team [9].

In the field of Spanish oncology, in the year 2018, the Andalusian Health Service approved the training of Advanced Practice Nurses (APNs) in oncological processes, due to the need to provide much more specialised nursing care. This role was created in the year 1971 in the USA, where it was implemented and defined [10]. An APN is a highly skilled nursing professional who has obtained advanced educational credentials and clinical training beyond the basic nursing education and licensing required of a registered nurse (RN). APNs are prepared through a postgraduate degree, such as a master’s or doctoral programme, which enable them to provide a higher level of care and take on roles that include direct patient care, consultation, education, research, and administration [11]. In the international context, advanced practice nursing is increasingly gaining momentum in different countries. However, there is a common denominator, which is the need to provide clarity about this role in the provision of health services [12]. In Spain, although there are certification systems and specific regulations at the regional level, this figure is neither certified nor regulated at the national level [10]. It is surprising that this role, which holds great responsibility in chemotherapy treatments and extravasation situations, remains unregulated in the country.

Therefore, the aim of this study was to analyse the state of the art on the role of nursing in the management of chemotherapy extravasation, recognising the possible risk factors and identifying effective training programmes for nurses.

## 2. Materials and Methods

In this study, a systematic review of the scientific literature was carried out, which gathered interventions related to the management of chemotherapy extravasation by nurses. This review collected aspects such as the role of nursing in the management of extravasation, the identification of risk factors, prevention strategies, and relevant actions in the case of extravasation. The PRISMA declaration criteria [13] for systematic reviews were applied, thereby thoroughly analysing the selected articles. This investigation is registered in the International Prospective Register of Systematic Reviews (PROSPERO) (ID: 512480).

The PICO format (problem/population, intervention, comparison, outcome) was used, formulated as follows:-Population: patients undergoing intravenous chemotherapy treatments.-Intervention: education, prevention, administration of intravenous chemotherapy, and management of extravasation.-Comparison: different chemotherapy procedures in which variables related to extravasation have been analyzed.-Outcome: studies that report evidence of risk factors, effective interventions, professional training programmes, and indicators of extravasation.

### 2.1. Selection Criteria

The systematic review was conducted between December 2023 and March 2024 in the Web of Science (WOS), Scopus, and PubMed databases, including those studies that were published in the last 9 years (between 2015 and 2024), and selecting only open-access articles.

### 2.2. Search Strategy

The keywords used to conduct the search were obtained from the MeSH descriptors, developed by the National Library of Medicine, and the thesauri of the descriptors used in Health Science (DeCS) were used. The Boolean operators AND and OR were employed.

For the search strategy, the following criteria were selected: for Scopus ((title–abstract–keywords (extravasación* or extravasation*), AND title–abstract–keywords (enfermería* or nursing*), AND title–abstract–keywords (quimioterapia* or chemotherapy*), AND title–abstract–keywords (tratamiento* or treatment*), and AND title–abstract–keywords (prevención* or prevention*)); for PubMed (extravasation* [title/abstract] or extravasacion* [title/abstract]), AND (chemotherapy* [title/abstract] or quimioterapia* [title/abstract]), AND (prevention* [title/abstract] or prevencion* [title/abstract]), AND (nursing* [title/abstract] or enfermería* [title/abstract]), and AND (treatment* [title/abstract] or tratamiento* [title/abstract]); and for WOS (extravasation* or extravasacion [topic]), AND (chemotherapy* or quimioterapia* [topic]), AND (prevention* or prevencion* [topic]), AND (nursing* or enfermería* [topic]), and AND (treatment* or tratamiento* [topic]).

### 2.3. Inclusion and Exclusion Criteria

The following inclusion criteria were applied: (a) studies with evidence of relevant results regarding the management of chemotherapy extravasation by nurses; (b) studies that identified risk factors for chemotherapy extravasation and the correct prevention and administration; (c) quantitative articles published in scientific journals; and (d) open-access articles. Therefore, only original studies were included.

This review excluded those studies that did not address the management of extravasation caused by parenteral chemotherapy administration, which comprised opinions, perspectives, systematic and bibliographic reviews, letters to editors, and commentaries. The suitability of the selected articles for the study object and the inclusion criteria, with the aim of increasing the reliability and accuracy of the process, was assessed by three of the authors of this study (A.A.-B., F.-J.G.-V., and J.-J.G.-I.). After reviewing the title, abstract, and keywords, the fourth author (D.M.-N.) resolved the discrepancies between these three authors and decided the inclusion or exclusion of the articles. This procedure was carried out manually in each of the databases. No record analysis software was used. The preselected articles were downloaded and, subsequently, all the authors carried out a more in-depth analysis of these studies by reading the full text.

The identification and selection of articles (both included and excluded) and the reason for their exclusion in the screening and selection phase is shown in the following flowchart (Figure 1), in compliance with the Preferred Reporting Items for Systematic reviews and Meta-Analyses (PRISMA) declaration, which was designed to improve the integrity of the report of systematic reviews and meta-analyses [14].

### 2.4. Data Extraction

The suitability of the selected articles for the aim and inclusion criteria of this study, with the aim of increasing the reliability and accuracy of the process, was assessed following the PRISMA declaration.

This process produced numerous judgements and actions after the search. Firstly, the title, abstract, method, results, and conclusions of each article were thoroughly reviewed, and the data were extracted as they were presented in their respective studies when they were reviewed.

This systematic review included variables according to the PICOS format (P: participants; I: interventions; C: comparisons; O: results; and S: study design) [15]. With this strategy, it was possible to delimit the inclusion criteria and, based on them, carry out a qualitative analysis of the results. Moreover, this study included other variables that were considered relevant: author, year of publication, country, article reference, study objectives, measurement variables, and scales.

After this analysis, it was possible to select, more accurately, the most suitable articles to be included in this work.

### 2.5. Presentation of the Results: Adherence to Quality Initiatives (PRISMA)

The presentation of the results of the primary studies, obtained through a systematic and reproducible methodology, was carried out qualitatively and quantitatively (Figure 1).

### 2.6. Quality Evaluation

The Effective Public Health Practice Project (EPHPP) [16] was used to carry out the quality analysis as well as for the selection of articles. This instrument provides an overall score for each study, evaluating six internal components (Table 1). A study with at least four strong internal components and no weak qualifications is considered strong. Those with less than four strong ratings and one weak rating were considered moderate. Lastly, those with two or more weak ratings were considered weak [16].

The conclusions of this analysis are shown in Table 1. Out of the nine articles included, one presented a strong global score [17], seven showed a moderate global score [18,19,20,21,22,23,24], and one obtained a weak global score [25]. It is important to detail that all articles presented strong scores in components such as the percentage of participants who reached the end of the intervention and data gathering. These are relevant components to be taken into account in this systematic review, since they are fundamental to the objective of study for this research. Therefore, strong scores in these internal components were decisive for the inclusion of articles in this review. In addition, none of the papers had a weak score on confounding factors.

**Table 1 healthcare-12-01456-t001:** Quality evaluation components and ratings of the EPHPP instrument.

COMPONENTS							
Articles	1	2	3	4	5	6	Global Score
Sharour [18]	M	W	M	M	S	S	M
Corbitt et al. [25]	M	W	M	W	S	S	W
Sivabalan and Upasani [19]	M	M	M	M	S	S	M
Larsen et al. [20]	M	S	M	M	S	S	M
Yu et al. [21]	M	M	M	M	S	S	M
Mohammed et al. [22]	M	M	M	W	S	S	M
Coyle et al. [17]	S	S	M	M	S	S	S
Lima and Silva et al. [23]	M	S	M	M	S	S	M
Mas et al. [24]	W	S	M	M	S	S	M

W = Weak; M = moderate; S = strong; 1 = risk of bias; 2 = design; 3 = confounding factors; 4 = masking; 5 = data gathering; and 6 = dropout.

## 3. Results

### 3.1. Selection of Studies and Data Extraction Process

In order to discard studies that did not meet the inclusion criteria and select those that might be relevant, after conducting searches, the abstract, title, and keywords of each article were reviewed.

A total of 23 studies were obtained after conducting searches and applying inclusion criteria. Firstly, the Web of Science (WOS) database was used, where four studies were identified; secondly, five studies were identified in Scopus; and finally, a total of fourteen articles were collected from PubMed.

A sample of 21 studies was obtained for full-text review after deleting 2 duplicate articles. In addition, 12 other studies were ruled out when eligibility criteria were applied. The reason for excluding these 12 articles from the systematic review was that, although in principle they met the inclusion criteria, after further reading, it was realilised that some of them did not meet the object of study (*n* = 9), others were systematic reviews (*n* = 3), or studies without intervention (*n* = 3), and the rest did not correspond to the characteristics of the population (*n* = 3). In order to reduce selection bias, each study was independently evaluated by three of the researchers in this paper (A.A.-B., F.-J.G.-V., and J.-J.G.-I.) to agree whether each document met the established criteria. If there was no consensus on the exclusion or inclusion of a study, a fourth researcher (D.M.-N.) mediated the decision.

### 3.2. Characteristics of the Studies: Results Synthesis

Table 2 presents the following information for each of the articles: participants, interventions, comparisons, results, type of study, year of publication, authors, variables and measuring instruments, country, and study objective.

Out of the nine articles included in this review, one (11.11%) was a cross-sectional descriptive study [18], two (22.22%) were quasi-experimental studies [22,25], two (22.22%) were experimental studies [17,19], one (11.11%) was a prospective cohort study [20], two (22.22%) were descriptive retrospective studies [21,24], and one (11.11%) was a diagnostic precision study [23].

Regarding the countries in which the different studies were carried out, two studies (22.22%) were performed in India [18,19], two (22.22%) in the USA [17,25], one (11.11%) in Australia [20], one (11.11%) in China [21], one (11.11%) in Egypt [22], one (11.11%) in Brazil [23], and one (11.11%) in France [24].

With regard to the participants, three studies (33.33%) analysed the results through the clinical evidence [18,22] or simulated evidence [25] of nurses, five studies (55.55%) drew conclusions from the participation of adult patients [17,19,21,22,23], and one study (11.11%) used pediatric patients [24].

**Table 2 healthcare-12-01456-t002:** Characteristics of the studies included in this systematic review.

Author/s; (Year); Country; [Citation]	Study Design	Comparisons	Study Objectives	Participants	Measurement Variables (Measurement Scales)	Interventions	Results/Conclusions
Sharour; (2020); India; [18]	Descriptive, cross-sectional study	A single experimental group composed of nursing professionals who worked in oncology	To evaluate the knowledge of oncology nurses about chemotherapy related to extravasation injuries, risk factors, preventive measures, and management practices	110 oncology nurses; 78 were men (70.9%) and 32 were women (29.1%)	Measured variables: information about chemotherapy extravasation. Injuries: risk factors, preventive measures, and management practices (for the gathering of this information, a questionnaire was developed and validated)	The participants were given a test with questions about the levels of comprehension, application, and analysis. The test included questions related to definition (2 questions), signs and symptoms (5 questions), risk factors (15 questions), preventive measures (13 questions), and management (15 questions)	The knowledge of the oncology nurses about the definition, signs, and symptoms of chemotherapy extravasation was satisfactory. However, the knowledge about risk factors was limited. In general, the results showed that a high percentage of the participants had correct information about the mastery of the procedure. Nevertheless, the study reported a deficit of knowledge about the place of insertion and the characteristics of the cannula. The results showed that there was also a deficit of knowledge among the participants with respect to all specific treatment practices. Continuous training, seminars, and workshops must be implemented for oncology nurses to increase their knowledge and strengthen their competencies
Corbitt et al.; (2017); USA; [25]	Quasi-experimental study	A single experimental group composed of nurses	To evaluate the efficacy of a training and simulation programme for the preparation, delivery, and administration of vincristine through mini bags to improve patient safety during chemotherapy administration	N = 93 nurses who attended simulation sessions	Observation of extravasation/complications (record sheet developed by authors). Evaluation of the simulation learning (validation questionnaire developed by authors)	Firstly, a multidisciplinary work group was created by pharmaceutical and nursing staff to plan safe simulation scenarios in the administration of chemotherapy treatments, as well as the protocols of collection, delivery, and administration of mini bags. Secondly, the training of nurses was carried out. Lastly, the procedure was performed through a simulation of the use of the mini-bag technique by trained nurses. At the end of the simulation, the learning and the technique were evaluated through a questionnaire and observation	The adoption of this process during the simulation helped to guarantee the safety and wellbeing of the patients. The authors concluded that the role of oncology nurses is important for guaranteeing the correct administration of the chemotherapeutic treatment and the prevention of injuries
Sivabalan and Upasani; (2016); India; [19]	Experimental study	Groups: - A control group of cancer patients received chemotherapy and routine care - An experimental group of cancer patients received chemotherapy and nursing interventions	To evaluate the effectiveness of the nursing interventions on the physical and psychological results in cancer patients under chemotherapy	N = 130 (65 control group and 65 experimental group)	Physical outcome variables: chemotherapy symptoms, pain, fatigue, oral mucositis, nausea and vomiting, and extravasation. Psychological outcome variables: emotional wellbeing, anxiety, depression, and worrying. For the gathering of the data, a previously validated semistructured interview programme was used	The patients of the experimental group received nursing care, such as venous access control, breathing exercises, and spiritual attention, whereas the control group only received the intravenous administration of chemotherapy. To gather the data, a previously validated semistructured interview programme was used. The health state of the patients was evaluated before initiating the chemotherapy and after the period of the interventions to gather information on the study variables	The cancer patients who received the nursing interventions obtained better mean scores after the test in chemotherapy symptoms, pain, fatigue, emotional wellbeing, anxiety, and depression than the patients who received the routine care
Larsen et al.; (2020); Autralia; [20]	Prospective cohort study	A single experimental group of adult patients hospitalised in two oncology units who required a peripheral intravenous catheter (PIV) for the treatment	To identify modifiable and non-modifiable risk factors of PIV failure in patients who required intravenous treatment for oncological and hematological affectations	N = 200	Place of catheter insertion; PIV size/caliber; anatomical location; insertion side; number of attempts needed (record sheet developed by authors); pain experienced (Likert scale developed by authors); primary dressing and additional fastenings; hydration state upon insertion (record sheet developed by authors); evaluation of phlebitis (Likert scale of pain and sensitivity); erythema; swelling; palpable cord; purulence; heat; and skin hardness (record sheet developed by authors)	A follow-up was conducted for the patients who had a PIV inserted. Standardised insertion policies were implemented during the study. Following this process, preparation of the skin occurred with 2% chlorhexidine gluconate (CHG) in alcohol at 70%. The PIVs were PIV BD Insyte™ Autoguard blood control catheters, and it was recommended to cover them with a bordered polyurethane dressing. After the interventions, the data were gathered	The failure rate of the PIV was 34.9%. They failed most frequently due to complications associated with occlusion or infiltration (n = 74, 18.7%), dislodgement (n = 33, 8.3%), and phlebitis (n = 30, 7.6%). Research on PIV has been widely focused on extravasation injuries in cancer patients; however, no cases of extravasation were reported in this study. Nevertheless, the conclusions are fundamental in the prevention of extravasation, since the demonstrated risk factors can lead to extravasation. Moreover, failure rates of PIV in patients under cancer treatment are high, and their sequelae may include delays in treatment and infection
Yu et al.; (2018); China; [21]	Retrospective study	A single experimental group of cancer patients under treatment with a totally implantable venous access port (TIVAP) device	To evaluate the rate and risk factors of late complications associated with the use of TIVAP in cancer patients, as well as to design nursing strategies to minimise late complications	N = 500 patients (177 men and 323 women)	Measured variables: information about late complications of TIVAP (catheter obstruction, infection, drug extravasation, and catheter exposure), as a function of sex, age, primary diagnosis, and duration of surgery and hospitalisation (the data were gathered through clinical records of the hospital and validated interviews developed by the authors)	After placing the TIVAP, the patients were followed up at 14 days and 1, 3, and 6 months after the insertion, through interviews via phone call to gather information about the study variables	Nineteen out of the five hundred patients developed late complications. Of these 19 patients, 14 presented catheter obstruction, 3 patients showed infection, 1 patient developed complications associated with drug extravasation, and 1 patient experienced catheter breakage. In general, there was a low rate of late complications with the use of TIVAP. Catheter obstruction was the most frequent late complication. The risk factors of late complications associated with TIVAP include age and certain cancers, such as breast, lung, and gastric cancer
Mohammed et al.; (2023); Egypt; [22]	Quasi-experimental study	A single experimental group composed of nurses who worked in the department of medical oncology	To evaluate the impact of a training programme on the performance of nurses in minimising chemotherapy extravasation	N = 40	Information about the knowledge of the nurses through knowledge questionnaires, with an observation checklist for nurses, and information about the rate of extravasation before and after the training programme (validated data-gathering instrument developed by authors)	A group of nurses received a theoretical and practical training programme of seven sessions to minimise chemotherapy extravasation. This training included chemotherapy contents, causes and risk factors of extravasation, signs, symptoms and their different levels, prevention and management, hand hygiene, insertion of peripheral catheters, safe administration, and management rules. The professionals were given a questionnaire before and after the training programme, to evaluate both their knowledge and the rate of extravasation in the patients before and after the nurses received the training	After the training programme, there was a decrease in the complaints of the patients in terms of extravasation, from 20% to 8%. Thus, this training improved the knowledge and performance of the nurses who received the programme, and the rate of extravasation decreased in the patients they attended
Coyle et al.; (2015); USA; [17]	Analytical-experimental study	A single experimental group composed of patients under peripheral vesicant agent treatment	To evaluate the result of a new chemotherapy infusion practice from a multidisciplinary approach	N = 21,600 infusion treatments (the number of patients who received the treatment is not specified)	Chemotherapy extravasation (valid clinical records)	Firstly, a literature and guideline review was carried out to establish a change of practice in chemotherapy infusion. This change of practice was addressed from nursing and medical teams. The practice included an initial and documented evaluation before the initiation of the treatment, documenting of implemented interventions, considerations about vascular access, monitoring intervals, and patient education. Once this new practice was agreed upon, it was applied to different patients for 6 months, and information about the study variables was gathered	The change of practice and policy resulted in a reduction of 90% in the peripheral administration of vesicant agents, with no extravasations being reported in the first 6 months of implementation
Lima and Silva et al.; (2023); Brazil; [23]	Diagnostic precision study	A single experimental group with patients under antineoplastic chemotherapy	To analyse the accuracy of the clinical indicators of vascular trauma in patients with antineoplastic chemotherapy	N = 200	Clinical data (chemotherapy treatment time and type of tumour) and clinical indicators of vascular trauma (vascular dysfunction), decrease in vascular blood flow, thrombus formation, hematoma, signs of infection in the catheter insertion place, delay in the infusion treatment, reduced vascular elasticity, deterioration of limb function, pain, and extravasation [validated questionnaire developed by authors]	Data were gathered through questionnaires that included sociodemographic information, clinical information, and indicators of vascular trauma in the participants	Confirmation of the presence of vascular trauma in patients under treatment, with a prevalence of 11%. Vascular elasticity, pain, and signs of infection were three clinical indicators identified as accurate, with high specificity. The precision of the clinical indicators may guarantee a safer identification of vascular trauma by nurses in patients under antineoplastic treatment. This identification would contribute to the planning of interventions, thereby reducing the costs and maximising the health outcomes of cancer patients
Mas et al.; (2020); France; [24]	Descriptive, retrospective study	A single experimental group composed of children who had received chemotherapy with vesicant drugs	To evaluate the result of the early saline wash procedure for extravasation caused by cytotoxic drugs in the upper limbs of children	N = 13	The aesthetic aspect, residual pain, and movement amplitude were analysed, as well as the time elapsed until surgery and the resumption of chemotherapy in the last follow-up (validated data-gathering instrument developed by authors)	Follow-up of 11 months, after drug extravasation (which was treated with saline wash), and gathering of data about the study variables	The saline wash procedure is safe and simple, and it significantly reduces the rate of extensive skin damage. Consequently, it also reduces tissue damage in possible cases of extravasation

### 3.3. The Role of Nursing in the Management of Extravasation

In relation to the role of nursing in extravasation, studies such as that of Coyle et al. [17] observed a direct implication with the necessary education that must be given to the patient who will start the chemotherapy treatment, both before and during the process, attaining a greater level of satisfaction. Sivabalan and Upasani [19] concluded that, in order for nurses to be able to provide an effective health education, they must receive adequate training in extravasation, with a solid foundation of knowledge, understanding, and attention competence, so as to provide comprehensive care to all cancer patients. The mentioned authors demonstrate that, through good preparation, the quality of life of patients and their families can be improved. They also identified that the nursing interventions reduced the physical symptoms of chemotherapy, such as pain and fatigue, as well as anxiety and depression, thereby helping to improve the emotional well-being of the patient. Similarly, Yu et al. [21] highlighted the importance of education strategies for patients with a totally implantable venous access port (TIVAP) implanted, as well as the relevance of the management and early identification of complications to prevent extravasation. Consequently, interventions performed by nurses are very positively accepted by patients, thus improving the response to the intervention.

Nursing is also involved in the action part in the cases of extravasation, as is reflected by Mas et al. [24], who report that extravasation is a surgical emergency that may have very severe sequelae. The mentioned study analysed the procedure of saline wash in children, reporting that this is a safe and easy process that reduced the rate of skin necrosis.

### 3.4. Risk Factors That Lead to Extravasation

One of the risk factors identified in the studies was the nurses’ lack of knowledge about them [18]. Deficient knowledge may affect the quality of the care that is given to patients who have suffered a case of extravasation. Moreover, the study of Lima and Silva et al. [23] identified three indicators of vascular trauma that, if not detected in time, could lead to extravasation. The most precise indicators were the decrease in vascular elasticity, pain, and signs of infection. Through these indicators and their early identification, it is possible to guarantee a safer detection of vascular trauma by nurses in chemotherapy patients, thereby contributing to the adequate planning of interventions and improving the health results of cancer patients. Another study [20] reported a high failure rate of peripheral intravenous (PIV) catheters, being more frequently due to complications associated with occlusion or infiltration, dislodgement, and phlebitis. All these complications can also lead to extravasation if they are not detected in time, and they may also produce sequelae such as treatment delay, thus worsening the quality of life of patients who receive cancer treatment. The strongest association found in this study was between the use of non-sterile fastening tapes and the significant decrease in dislodgement. Lastly, Yu et al. [21] observed that catheter obstruction is the latest complication associated with the central venous catheter (TIVAP), with age and certain cancers (breast, lung, and gastric) being the main risk factors.

### 3.5. Effective Training Programmes for Nurses

Mohammed et al. [22] reported the great efficacy of the training programme that was conducted with the nursing staff, including contents of chemotherapy, risk factors, and prevention. A questionnaire was administered before and after the training to evaluate the learning, obtaining very satisfactory results, in both the nurses and the rate of extravasation, where the complaints from the patients decreased from 20% to 8% after the implementation of the programme. Corbitt et al. [25] drew similar conclusions in their study, where, through the simulation of vincristine administration via a mini bag and the training of the nurses, the safety and well-being of the patients were improved, also obtaining very promising results with regard to the prevention of extravasation.

## 4. Discussion

The general aim of this systematic review was to analyse the state of the art on the role of nursing in the management of chemotherapy extravasation, recognising the possible risk factors and identifying effective training programmes for nurses. All key aspects of the discussion of results will be addressed below.

### 4.1. Risk Factors for Chemotherapy Extravasation

Sharour [18] demonstrated that oncology nurses had adequate knowledge about the signs and symptoms of extravasation, and deficient knowledge in terms of specific treatments, characteristics of cannulas, and insertion areas. These findings are in line with the conclusions of Kosgeroglu et al. [26], who observed the same deficits of knowledge in nurses. Another study [27] reported that, in contrast with the findings of Sharour [18], the nurses presented poor knowledge regarding the identification of signs, symptoms, and risk factors. With these conclusions, it could be argued that a relevant variable that leads to extravasation is the nurse’s deficient knowledge about different aspects related to chemotherapy treatment [28].

Furthermore, with regard to the identification of risk factors of extravasation, Lima and Silva et al. [23] observed that vascular trauma is an iatrogenic event that becomes more prevalent along the years. The identified indicators were the following factors: extravasation, pain, and changes in skin colour. These findings are in agreement with those of similar studies [29,30] that detected a decrease in elasticity, pain, and signs of infection in the catheter puncture area, with high specificity values. Nurses play an essential role in the care of patients under antineoplastic chemotherapy, and they must manage possible adverse events to prevent extravasation and toxicity. In this context, the early identification of vascular trauma through clinical indicators becomes a fundamental task, since three accurate clinical indicators are enough to predict vascular trauma and the subsequent extravasation, thereby improving patients’ care quality [31,32].

### 4.2. Nursing Procedures and Interventions in Chemotherapy Treatment

Sivabalan and Upasani [19] reported that the patients who received chemotherapy suffered multiple physical and psychological symptoms, leading to poorer general well-being. In this sense, Miaskowski et al. [33] observed that most cancer patients suffer from pain (80%), which is the most frequent symptom. Other very common symptoms observed in this population are fatigue and changes in appetite [19,34]. Other studies, such as that of Jadoon et al. [35], also found that over half of these patients suffered from depression and anxiety. These situations can be attenuated with the implementation of nursing interventions, since adequate planning of the latter can improve the physical and emotional well-being of the patients [19,36,37,38,39,40,41]. The symptoms related to the psychological state, such as anxiety, depression, and emotional well-being, may significantly improve through nursing interventions, e.g., massages, progressive muscle relaxation, breathing exercises, psychoeducational attention, and praying [42,43].

The findings of Larsen et al. [20], which are in agreement with those of other studies, show that adverse consequences lie especially in complications related to occlusion and infiltration [44,45]. In this line, other studies identified a series of risk factors, such as the application of non-sterile fastening tape, secondary to the primary dressing as a protective factor [46,47], or two or more attempts to reinsert the catheter [48]. It is necessary to carry out improvements in the insertion, care, maintenance, and extraction of PIV in cancer patients, in order to guarantee the health and preservation of the blood vessels in the long term, thereby complying with the practice standards for access devices in oncology nursing [49,50].

The study conducted by Yu et al. [21] showed, in general, a low rate of late complications with the use of the central venous catheter (TIVAP) in cancer treatment. This rate is similar to that demonstrated in other studies, which also found other adverse effects such as infection [51]. However, other articles report higher rates [52,53]. Other authors [54] agree with Yu et al. [21] in terms of age and certain underlying diseases, which were associated with a greater risk of late complications. Nevertheless, the rate of infection in this study was much lower than the previously reported rate [55]. In the latter work, there was only one case of extravasation, with symptoms such as local subcutaneous swelling, pain, and heat. This complication was resolved with hydropathic dressings and treating the obstruction. The findings of this work, regarding the selected intervention in cases of catheter exposure, are in line with those of other studies [56], where the most effective treatment of choice was the removal of the catheter through surgical intervention. The conclusions drawn in these studies underline the importance of the management of TIVAP being carried out by certified nurses, with the aim of minimising the possibilities of drug extravasation [57].

In relation to the consequences of extravasation, Mas et al. [24] concluded that extravasation injuries may cause very severe and permanent lesions, such as skin necrosis, skin infections with abscesses, and necrotising fasciitis. In this sense, other studies showed similar findings, as well as sequelae such as antiesthetic skin, muscle contractions, and finger amputation [58,59]. All of these findings further support the importance of preventing chemotherapy extravasation as a point of public health improvement, since, on the one hand, it represents economic savings in the care that future extravasation lesions may require and, on the other hand, the quality of life of cancer patients and the efficiency of health care are improved [6,60].

### 4.3. Training of Nursing Professionals in Chemotherapy Treatments

With regard to the training of nurses in chemotherapy extravasation, the study of Mohammed et al. [22] applied a training programme and analysed its results. Some of their findings are in agreement with the results of similar works [61]. The results obtained after applying the programme showed an increase in knowledge about reducing extravasation. In this sense, similar studies also show improvements in the clinical practice scores of the nurses after the application of the programme [62,63]. Furthermore, another study [25] also identified an improvement in the professional competences regarding the prevention of extravasation after the application of a simulation-based practice training. Similarly, multidisciplinary training and intervention have shown substantial improvements in the management and prevention of extravasation; this finding has been reported in two of the articles analysed in this systematic review [17,25]. These conclusions support the training of nursing staff in this topic for the prevention of extravasation and the improvement of patient safety [28].

### 4.4. Limitations and Future Lines of Research

Lastly, it is important to highlight that this study identified an important volume of relevant content for addressing chemotherapy extravasation. However, it also presents a series of limitations. Firstly, it was difficult to find a great diversity of studies about the study topic in the databases. This could be due to the fact that it is a study with poorly analysed objectives and novel inclusion criteria, of which there is little information. Moreover, the findings show very general conclusions; thus, in order to carry out intervention propositions aimed at more specific groups, it is necessary to also incorporate the evidence of other studies about specific groups, since this study is not focused on a single population group. Finally, it should be noted that this systematic review is a pilot study, from which the study researchers have obtained basic information about the study topic. In the future other more specific areas of the topic will be addressed. Because this is a pilot study, only open access works were included in this first review to obtain an overview. Furthermore, to further refine the searches in more specific areas of nursing, it would be convenient to also add the CINHAL database.

Therefore, future studies should propose new systematic reviews adapted to differential factors of a specific population, with a specific type of cancer and specific chemotherapeutic treatments for that type of cancer, taking into consideration a greater margin of years regarding the publication date for database searches, thus obtaining the findings needed for planning general and specific interventions with a scientific background. It is also important to carry out a meta-analysis on future systematic reviews, in order to obtain significant results. In addition, implementation studies/quality improvement to ensure translation into practice would also be appropriate.

### 4.5. Implications for the Clinical Practice

The findings shown in this study provide valuable data for clinical practice. Among the most relevant implications, it is worth highlighting the following:∘Assessment of the importance of comprehensive cancer patient care;∘Increase in the safety of the cancer patient and improvement of their quality of life;∘Promotion of interdisciplinary and multidisciplinary work;∘Improvement of the professional competence of nurses;∘Favouring of the specialisation within the nursing profession.

## 5. Conclusions

Clinical and care procedures must be continuously evaluated to identify strong and weak points, as well as aspects to be improved. This study delved into the role of nursing in chemotherapy treatment extravasation. It is a very important topic that requires constant updating to improve the quality of life of the patient. The most relevant conclusions and findings identified in this systematic review are:Nursing plays an important role in the management, prevention, and treatment of chemotherapy extravasation in cancer patients. These are professionals who are a fundamental piece in the multidisciplinary team that provides comprehensive care to the patient, thus attending to patients’ biopsychosocial health. Through nursing interventions, different health education strategies are also applied. With these interventions, both the patients and their families can rapidly detect complications in the treatment, and thus further improve the safety of the patient;It is necessary for nurses to have up-to-date knowledge about this topic. Adequate training improves the competence level, efficiently shows the risk factors, prevents a larger number of extravasations, improves the health results, and reduces the costs of cancer patients;The quality of life of cancer patients must be considered a relevant aspect from the perspective of public health. In this sense, the topic analysed in this study must be addressed from multiple agents and levels, such as healthcare, community, political, and social.

To sum up, this study shows the way to improve the care provided to the oncological patient, in general, and the prevention of extravasation, in particular. This information can be used as a starting point to develop training and intervention propositions, and thus to improve the quality of life of cancer patients and their families. This systematic review also contributes to the empowerment of nurses through the promotion of their professional competence in this area.

## Figures and Tables

**Figure 1 healthcare-12-01456-f001:**
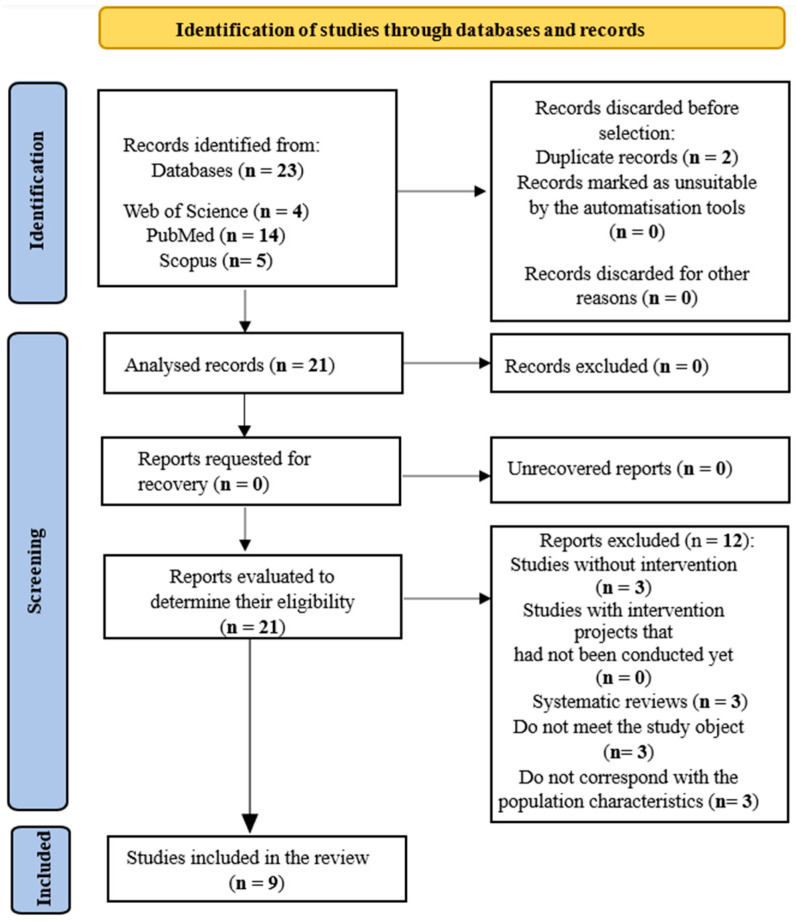
Flowchart of the systematic review process according to the declarations of the PRISMA protocol.

## Data Availability

Further data that support the findings of this study are available upon reasonable request from the corresponding author. Some data are not publicly available due to privacy or ethical restrictions.

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
