# Peer review of "The Role of Nursing in the Management of Chemotherapy Extravasation: A Systematic Review Regarding Public Health"

_healthcare, 2024, doi:10.3390/healthcare12141456_

Round 1

Reviewer 1 Report (Previous Reviewer 2)

Comments and Suggestions for Authors

Thanks to the author for his hard work in revising the work. However, the manuscript requires minor editing by a native English speaker.

Comments on the Quality of English Language

 Minor editing of English language required

Author Response

Dear reviewer,

Thank you very much for the recommendations.

Document with answers is attached.

Kind regards,
The authors.

Reviewer 2 Report (Previous Reviewer 3)

Comments and Suggestions for Authors

Dear Authors,

I appreciate your efforts to improve the quality of this manuscript, however many suggestions have not been properly integrated and the current version is less effective than the previous one. Again, I would like to make some suggestions for improving the quality of your manuscript:

-Abstract. The abstract must be discursive, so a review of the structure is necessary. Delete the chapters name (e.g. background, methods, results, ...) and if possible show the lists in discursive and linear form (eliminate: "Among the most relevant findings are: the ..."; use "Among the most relevant findings are the ...".

- Introduction, line 35. This sentence is not well formulated. Please, reparfrasate the text making the concept understandable.

- Introduction, lines 64-66. This period in unclear. Please, explain the meaning and if necessary change the description in the text.

- Introduction, lines 70-85. It is not clear the relationship between APN and your aim, and honestly I’m not going because you should talk about APN at this point. The focus of your goal is another.

- Methods, lines 97-103. The PICO is not a question, and it is still unclear. I suggest to you to review this section and if unable, to avoid its representation.

- Methods, lines 113-123. The spelling errors were not corrected (e.g. Tittle/Abstract). Why did you use spanish language in the string if you declare that no language filters were applied? It is not methodogically correct. Inclusion criteria are non analytical and linear. Please, redefine them (specifically, a and c are redundant and not clear). Line 127, I see additional typo. Please check the quality of the manuscript before resubmitting.

- Methods, lines 137-140. It is still necessary to declare who did what and with what tools. Did you use a software for the analysis of records? Please specify.

- You declared: "The table format has been improved by redistributing some columns". Currently, it is difficult to read. The use of the closing point is either always necessary or must be removed, the formatting must be set correctly. The text must be aligned correctly. Acronyms are not correctly presented. What is PIV BD? Use the correct width for the columns to ensure a simple and linear readability. A reader cannot strive to find information.

Author Response

Dear reviewer,

Thank you very much for the recommendations.

Document with answers is attached.

Kind regards,
The authors.

Reviewer 3 Report (Previous Reviewer 4)

Comments and Suggestions for Authors

Dear authors:

I have reviewed your paper entitled “The role of nursing in the management of chemotherapy extravasation: a systematic review with regard to public health, for a second time. And I want to congratulate your effort to improve it. I believe it, absolutely, better. I just suggest adding at limitation point, the fact you did not use CINHAL data base (which is specific from nursing, and being a paper about nurses, would be interesting use it).

And in title reflect about changing “ with regard” by “regarding”.

I have nothing to add, and I wish you good luck towards publishing it!

Best regard.

Author Response

Dear reviewer,

Thank you very much for the recommendations.

Document with answers is attached.

Kind regards,
The authors.

This manuscript is a resubmission of an earlier submission. The following is a list of the peer review reports and author responses from that submission.

Round 1

Reviewer 1 Report

Comments and Suggestions for Authors

I enjoyed your manuscript submission regarding a systematic review of nursing role in management of chemotherapy extravasation. Strong systematic review methods description. 

Please comment on potential issues and/or rationale for only including open-access articles. Needs to be listed as a potential limitation in discussion. However, the care recommendations related to this have not significantly changed in past 9 years. Table 2 could be moved to Supplemental file if the journal does not have pages for this detailed information.

Please do not include repetition of findings in discussion but focus on recommendations for practice (i.e. Risk factors for extravasation; identifying extravasation, management and prevision; chemotherapy administration training programs for all nurses who administer chemotherapy, in particular vesicant agents). Please also include your recommendations for future research, which might include implementation studies/Quality Improvement to ensure translation into practice. I look forward to seeing a future revision of this important work.

Author Response

Dear reviewer,

Thank you very much for the recommendations.

Document with answers is attached.

kind regards,

The authors.

Reviewer 2 Report

Comments and Suggestions for Authors

1.Please have a native English speaker edit the entire article to enhance readability and ensure correct tense usage. Include proof of editing.

2. Mention the PROSPERO registration number in the abstract.

3. On LINE 67, explain what "EPA" stands for and provide the full name the first time it appears.

4.           4.On LINES 60-61, are these percentage figures attributed to nursing staff?

5.5.On LINE 85, the PROSPERO ID seems incorrect as it usually follows the format CRD+11 digits; please clarify.

6.6.On LINE 97, why include only open-access articles? Excluding articles that meet your PICO criteria could introduce bias in your results. The study targets nursing staff, yet the search strategy, such as the "P" in PICO, does not clearly relate to nursing staff.

7.7.The formatting of Table 1 lacks clarity and consistency; for instance, abbreviations like PIV, BD, TIVAP should be explained in the notes. The Interventions and Results sections should be concise and to the point.

8.8.The discussion seems to jump between different studies and topics without a clear structure. It might be more effective to organize the content into sub-sections, each focusing on a specific aspect of nursing management such as knowledge, interventions, and outcomes.

9.9.Since the manuscript title mentions "with regard to public health," there should be a clear discussion on how effective management of chemotherapy extravasation by nurses impacts public health outcomes. This might include considerations of healthcare costs, patient quality of life, and broader healthcare system efficiencies.

Comments on the Quality of English Language

Extensive editing of English language required

Author Response

(The authors gave the same response as above.)

Reviewer 3 Report

Comments and Suggestions for Authors

Dear Authors, your manuscript offers an important source of knowledge regarding the nursing role in the management of chemoterapy extravasation. I give you some suggestions to improve the scientific sound of your work:

- Introduction (line 33). Please modify the initial part of the period. The word "historically" is not correct if we think to demographic and epidemiological transitions.

- Materials and Methods (line 84). How do you apply the Declaration of Helsinki to a systematic review? Please specify it.

- Materials and Methods (line 87-93). The PICO is uncorrect. Please, modify it.

- Materials and Methods (line 99). It is not clear if you used the same research string for the different databases. Please specify it and verify if the string is correct/updated. There are also many spelling errors in the given string e.g. Tittle. Article selection filters and language criteria are not indicated.

- Materials and Methods (line 115). The inclusion criteria are not clear. In particular (a), (b) and (c). Please, clarify them.

- Materials and Methods (line 123). It is not clear how did you manage the abstract/full text and if you used specific software. The steps of the process are not clear. Please, specify them.

-Table 2. I suggest to you to review the content of this table for the formatting, consistency and presence of typos.

Author Response

(The authors gave the same response as above.)

Reviewer 4 Report

Comments and Suggestions for Authors

Dear authors:

I have reviewed your paper entitled “The role of nursing in the management of chemotherapy extravasation: a systematic review with regard to public health. It was made a systematic review registered in PROSPERO, that aim to analyses the role of nurses in the management of chemotherapy extravasation, recognizing the risk factors and identifying effective training programs for the nursing staff. The results show that the management of chemotherapy extravasation is closely related to factors that largely depend on the nursing staff.

Chemotherapy extravasation is a complication that delay patients’ treatments, demanding a prevention actions and effective nursing training, to deal and prevent that situation. This study is important to implement in future, effective practices to prevent these occurrences. So, in my opinion, this study is considered important and relevant, so congratulations for your effort and contribution in the area. I believe the readers of the journal will read it with the same interest I did.

I have some suggests to reflection and improve your paper:

 - In abstract and pages 2 and 18: the aim should be written in the same way.

-In abstract the results are very incomplete, do not give answer to objectives.

- Page 1 line 37-38 – change “280,110” to “280.110” and “113,000” to “113.000”;

- In materials and methods: line 83-84 “The study was performed in 83

-In page 2 line 83-84: you wrote "compliance with the guidelines of the Declaration of Helsinki.”. Being a systematic review what you mean with this sentence?

- Could you give a link in Prospero registration (not only the ID)?

-Page 3: why did you limited time (last nine years)? The last review about the area was published in that year?!

-Why did you not use CINHAL data base (specific data base for nursing, when your focus was on “role of nursing”? (limitation)

-Why you use as filter only to different language? (Limitation);

-Page 5 line 187 (change N by n);

- Results and discussion should be more focus on the objectives lined;

- Discussion: should be insert a paragraph with limitations to the study (i.e. the critical appraisal to assess quality of the articles could be used the equator checklists);

I have nothing to add, and I wish you good luck towards publishing it!

Best regards.

Author Response

(The authors gave the same response as above.)
